# Seasonal Single-Site Sampling Reveals Large Diversity of Marine Algal Toxins in Coastal Waters and Shellfish of New Caledonia (Southwestern Pacific)

**DOI:** 10.3390/toxins15110642

**Published:** 2023-11-03

**Authors:** Manoëlla Sibat, Tepoerau Mai, Simon Tanniou, Isabelle Biegala, Philipp Hess, Thierry Jauffrais

**Affiliations:** 1Ifremer, ODE/PHYTOX/METALG, Rue de l’île d’Yeu, F-44300 Nantes, France; simon.tanniou@ifremer.fr; 2Ifremer, IRD, Univ Nouvelle-Calédonie, Univ La Réunion, CNRS, UMR 9220 ENTROPIE, 98800 Nouméa, New Caledonia; tmai@ilm.pf (T.M.); thierry.jauffrais@ifremer.fr (T.J.); 3Institut Louis Malardé (ILM), 98713 Papeete, Tahiti, French Polynesia; 4Aix Marseille Univ, Université de Toulon, CNRS, IRD, MIO, UM110, 13288 Marseille, France; isabelle.biegala@ird.fr

**Keywords:** passive sampling, solid-phase adsorption toxin tracking, shellfish, mass spectrometry, New Caledonia, cyclic imine, diarrheic shellfish poisoning, azaspiracid, brevetoxin

## Abstract

Algal toxins pose a serious threat to human and coastal ecosystem health, even if their potential impacts are poorly documented in New Caledonia (NC). In this survey, bivalves and seawater (concentrated through passive samplers) from bays surrounding Noumea, NC, collected during the warm and cold seasons were analyzed for algal toxins using a multi-toxin screening approach. Several groups of marine microalgal toxins were detected for the first time in NC. Okadaic acid (OA), azaspiracid-2 (AZA2), pectenotoxin-2 (PTX2), pinnatoxin-G (PnTX-G), and homo-yessotoxin (homo-YTX) were detected in seawater at higher levels during the summer. A more diversified toxin profile was found in shellfish with brevetoxin-3 (BTX3), gymnodimine-A (GYM-A), and 13-desmethyl spirolide-C (SPX1), being confirmed in addition to the five toxin groups also found in seawater. Diarrhetic and neurotoxic toxins did not exceed regulatory limits, but PnTX-G was present at up to the limit of the threshold recommended by the French Food Safety Authority (ANSES, 23 μg kg^−1^). In the present study, internationally regulated toxins of the AZA-, BTX-, and OA-groups by the Codex Alimentarius were detected in addition to five emerging toxin groups, indicating that algal toxins pose a potential risk for the consumers in NC or shellfish export.

## 1. Introduction

The consumption of seafood contaminated with algal toxins can lead to varying types and severity of food poisoning, as well as other diseases, such as dermatitis and respiratory illness [1]. In the marine environment, such toxins are mainly produced by Dinophyceae (dinoflagellates) and Bacillariophyceae (diatoms), but also by Prymnesiophyceae, Raphidophyceae, and Cyanophyceae (cyanobacteria) [2]. These toxins are classified based on their chemical structure and lipophilicity or by the symptoms they provoke in humans (Table 1). The syndromes associated with these toxins include paralytic shellfish poisoning (PSP), diarrheic shellfish poisoning (DSP), amnesic shellfish poisoning (ASP), neurotoxic shellfish poisoning (NSP), as well as other conditions that are not yet well characterized. The consumption of seafood contaminated by phycotoxins has led to the implementation of protective measures in many countries [3]. In Europe, regulatory limits for some groups of phycotoxins have been implemented following the European regulation N°853/2004 [4]. When threshold values are reached or exceeded, shellfish production or recreational fishing areas are closed and sale is prohibited to prevent human illness and ensure public health. These closures of production areas also have a direct impact on shellfish producers as they cause economic losses. In addition, they indirectly affect the sector by presenting a negative image to consumers and large retailers. 

Furthermore, like other islands in the Southwestern Pacific, the coastal ecosystems in New Caledonia (NC) currently face multiple pressures including global change, mining, aquaculture production, tourism, and urban and farming pollution [5]. These factors can induce algal blooms with adverse effects on coastal ecosystems or human health [6,7,8]. Indeed, harmful algal blooms (HABs) have increasingly been reported globally in recent decades [9]. This perceived increase may be associated with naturally occurring environmental changes [10,11] and/or local anthropogenic influences [9]. In NC, some studies have started to investigate the potential impact of HABs, including green tide species, *Ulva* spp. [6,8,12], on cyanobacterial blooms (e.g., *Lyngbya*) associated with acute cases of dermatitis [7,13] or related to ciguatera [14,15] and the potential impact of harmful algae on shrimp culture [16]. 

Tourism, reef and lagoon fisheries, and aquaculture are major economic activities in NC. Shrimp aquaculture started in the 70s and has a production of approximately 1500 tons each year, covering the local market demand and allowing the export of blue shrimps (*Penaeus stylirostris*) to other countries, notably Japan. In addition, approximately 4.5 to 15 tons of bivalve mollusks are taken per year by professional fishers [17,18,19] and two oyster farms exist for the local market. Additionally, pilot studies on Pectinidae (*Amusium balloti*) fishing are being performed in the northern province in partnership with Australia [17]. 

During HABs, marine biotoxins can accumulate in shellfish and pose a significant safety risk to consumers. Since the 2010’s, liquid chromatography–tandem mass spectrometry (LC-MS/MS) analyses have been carried out in Europe as the official testing method for marine biotoxins to ensure public safety [20]. Many countries run monitoring programs and have implemented directives and legislation for marine biotoxins to minimize the risk to the population. As well as the economic impact (e.g., Japan [21], the US-FDA [22], and Australia/New Zealand [23]). However, the potential impacts of harmful algae and associated phycotoxins are still poorly documented in this region of the Southwestern Pacific. There is thus a lack of knowledge regarding potential sanitary and socioeconomic threats to seafood industries and/or tourism activities. 

Most monitoring programs rely on both phytoplankton monitoring through the identification of potentially toxic species and on seafood monitoring through the identification and quantification of regulated toxins [24]. However, these methods have some limitations, e.g., phytoplankton monitoring is labor-intensive, “only” presents a snapshot of the diversity at a given time and location, requires taxonomic expertise, and may be inconclusive for small toxic species (<10 to 20 µm) or in places where knowledge of harmful algae diversity is still scarce [25,26]. As the identification and quantification of toxins in seafood is a prerequisite for food safety, owing to the biotransformation of phytoplankton toxins in filter feeding bivalves or herbivorous fishes, analysis using passive samplers is a good alternative for screening phycotoxins dissolved in water [27]. This is why the use of passive samplers was developed for monitoring and/or identifying marine toxins during algal blooms as significant amounts of dissolved toxins are released into the seawater, including-low polarity lipophilic compounds [27]. MacKenzie et al. (2004) identified a passive sampler suitable for the accumulation of lipophilic toxins, namely, solid-phase adsorption toxin tracking (SPATT), and this method was then used, tested, and found to be highly efficient in different parts of the world on a wide variety of marine lipophilic toxins [26,28,29,30,31,32].

NC, as a territorial collectivity of the French Republic, is not required to follow European regulations on phycotoxins; however, for the protection of local consumers or for exports, it would be appropriate to follow international standards such as the consensus-based regulation by *Codex alimentarius* [3]. However, monitoring systems are costly to implement and need to be based on risk likelihood and gravity. As the occurrence of most toxins in NC is not known, the present study aimed at investigating the presence of major known toxins, including both internationally regulated and emerging toxin groups. A dual approach was selected, sampling both sea water, using the concentration of dissolved toxins by passive samplers (SPATTs), and filter-feeding bivalves. Passive samplers were deployed for one week in Lemon Bay, one of the bays surrounding Noumea on the southwestern coast of NC, during both the warm and cold seasons. Filter-feeding bivalves were sampled during the same week in the same area, as well as in two adjacent bays. 

**Table 1 toxins-15-00642-t001:** The main marine algal toxin groups, their primary sources, lipophilicity, syndromes they cause and their regulatory limit (**in bold**) or recommended threshold value.

Toxin Group	Species Involved	Lipophilicity	Syndrome	Regulatory Limit (in Bold) or Recommended Threshold Value
Azaspiracids	*Azadinium* spp.	Lipophilic	DSP ^1^	**160 µg AZA1 eq. kg^−1^ of total weight** [4,33]
Brevetoxins	*Karenia brevis*, *Chatonella* spp.	Lipophilic	NSP ^2^	**800 µg BTX2 kg^−1^ of total weight** [3,34]
Ciguatoxins	*Gambierdiscus* spp., *Fukuyoa* spp.	Lipophilic	NSP, ciguatera poisoning	0.01 µg CTX1B eq. kg^−1^ of total weight [34]
Cyanotoxins	*Hydrocoleum* sp., *Oscillatoria* sp., *Anabaena* sp., *Lyngbya* sp.	Hydrophilic	Neurotoxic, dermatotoxic, and hepatotoxic	0.14 µg toxin day^−1^ kg^−1^ of body weight [35,36]
Domoic Acid	*Pseudo-nitzchia* spp.	Hydrophilic	ASP ^3^	**20 mg DA kg^−1^ of total weight**
Gymnodimines	*Karenia selliformis*, *Alexandrium ostenfeldii*, *Alexandrium peruvianum*	Lipophilic	Not yet well characterized, but toxic to mice	N/a
Maitotoxins	*Gambierdiscus* spp., *Fukuyoa* spp.	Amphiphilic	NSP	N/a
Microcystins	*Mycrocystis* spp., *Planktothrix* spp.,	Lipophilic	Neurotoxic and hepatotoxic	1 ng MCs day^−1^ kg^−1^of body weight [35,37]
Okadaic Acid	*Dinophysis* spp., *Prorocentrum* spp., *Phalacroma* spp.	Lipophilic	DSP	**160 µg OA eq. kg^−1^ of total weight** [3,4]
Palytoxins	*Ostreopsis* spp	Amphiphilic	Respiratory and dermatotoxicity	30 µg PLTX kg^−1^ of total weight [38]
Pectenotoxins	*Dinophysis* spp. *Prorocentrum* spp.	Lipophilic	Not yet well characterized, but cardiotoxic, hepatotoxic, and neurotoxic in animal models	deregulated
Pinnatoxins	*Pinna attenuata* *Vulcanodinium rugosum*	Lipophilic	Not yet well characterized, but highly toxic to mice	23 µg PnTX-G kg^−1^ of total weight [39]
Pteriatoxins	*Pteria penguin*	Lipophilic	Potent toxicity	N/a
Saxitoxins	*Alexandrium* spp., *Gymnodinium catenatum*	Hydrophilic	PSP ^4^	**800 µg STX kg^−1^ of total weight** [3,4]
Spirolides	*Alexandrium ostenfeldii,* *Alexandrium peruvianum*	Lipophilic	Not yet well characterized but toxic to mice	N/a
Tetrodotoxins	*Tetraodontinae*	Hydrophilic	TTX poisoning (PSP-like)	44 µg TTX kg^−1^ of total weight [40]
Yessotoxins	*Protoceratium reticulatum*, *lingulodinium polyedrum*, *Gonyaulax spinifera*	Lipophilic	Not yet well characterized, but cardiotoxic, hepatotoxic, and neurotoxic in animal models	**3.75 mg YTX eq kg^−1^ of total weight** [3,4]

^1^ DSP: diarrheic shellfish poisoning, ^2^ NSP: neurotoxic shellfish poisoning, ^3^ ASP: amnesic shellfish poisoning, ^4^ PSP paralytic shellfish poisoning. N/a: not applicable (no threshold value known).

## 2. Results

All samples were analyzed using LC-MS/MS based on a multi-toxin screening approach, leading to the detection of major lipophilic groups of marine biotoxins. The toxins profile observed (Figure 1b–f and Figure 2) in the two types of samples (SPATTs and shellfish) belonged to the group of regulated lipophilic toxins (AZAs, BTXs, and OA), toxins only regulated in the EU (YTXs), and the PTX-group, as well as the cyclic imines group (PnTXs, GYMs, and SPXs). No MC, PLTX-like, and CTX, toxins produced by *Gambierdiscus* or *Fukuyoa* genera were detected in the samples. The profile of the shellfish was more diversified depending on the species analyzed, but they all contained the cyclic imines GYM-A and PnTX-G (Figure 1c,f and Figure 3).

### 2.1. SPATTs Samples

The results showed good repeatability between replicates for both the warm and cold seasons, with a coefficient of variation CV < 25% (Appendix A). The toxin profile was consistent between the two seasons with the presence of OA, homo YTX, AZA2, PnTX-G, and PTX2 (Figure 2), but the concentration of toxins was twice higher for AZA2 and 20 times higher for PTX2 during the warm season than during the cold season. An exception was observed for OA and homo-YTX, where the concentrations remained similar, with OA at around 37.2 ng SPATT^−1^ and homo YTX at around 14.6 ng SPATT^−1^ regardless of the sampling period. 

### 2.2. Shellfish Samples

The toxin profiles of shellfish samples were species-specific; however, they all contained the cyclic imines GYM-A and PnTX-G (Figure 3). As in the SPATTs, AZA2 was found in very small quantities (<1.2 µg kg^−1^ of WT) in three samples (*Pteria straminea*, *Modiolus* cf. *auriculatus*, and in one of the three samples of *Barbatia foliata*). The quantitative results (Appendix A) revealed a more diversified toxin profile for two species, *Pteria straminae* (WT N°1) and *Modiolus* cf. *auriculatus* (WT N°4), with totals of seven and six toxin groups detected, respectively. *Pteria straminea* (WT N°1) mostly contained YTXs (9.3 µg eq. YTX kg^−1^ of WT), composed of three analogues (Homo YTX, Homo COOH YTX, and Homo 45-OH YTX); mussels (*Modiolus* cf. *auriculatus*, WT N°4) contained OA, homo-YTX, PTX2sa, AZA2, GYMA, and PnTX-G, but at low levels (≤7 µg kg^−1^ of WT); while the giant clam (*Tridacna maxima,* WT N°3) was the only species where PTX2 was detected in addition to its metabolization product, PTX2sa. The three samples of *Barbatia foliata* bivalves (WT N°5 to WT N°7) collected from the same site (Lemon Bay, Noumea) had different profiles and concentrations of toxins. In addition to having accumulated several groups of toxins (OA, PTX2sa, AZA2, and GYMA), sample WT N°7 (*Barbatia foliata*) showed a high concentration of PnTX-G 22.6 µg kg^−1^ of whole flesh tissue. The lowest toxin content (≤0.6 µg kg^−1^ of WT) was detected in the oysters *Pinctada margaritifera* (WT N°8) collected in Ouemo (Sainte Marie Bay). The two samples of bivalves collected at Vata Bay had an identical profile with a quantity of GYM A four times greater than that of PnTX-G (WT N°9 and 10, *Anadara* sp., *Isognomon isognomon*, respectively).

## 3. Discussion

The present study permitted: (i) comparing toxin accumulation in SPATTs during two seasons (warm and cold), (ii) comparing the toxin profiles accumulated in different shellfish species and in SPATTs deployed from which the shellfish had been sampled (August 2021 and February 2022), and (iii) establishing a first inventory of the potential impact of lipophilic phycotoxins on sanitary and socioeconomic threats on seafood in NC.

### 3.1. SPATTs and Seasonality

SPATT sampling is a sensitive method and, owing to the high adsorption capacity of the HP20 polymeric resin, it is accepted and now commonly used to detect dissolved toxins in seawater from either pelagic or benthic phytoplankton [27]. In 2021, SPATTs were deployed in Lemon Bay for one week in August (cold and dry period in the southern hemisphere) and in 2022, during February (warm and wet period). For each season, three SPATTs were deployed, which allowed the detection of OA, PTX2, homo-YTX, PnTX-G, and AZA 2. The levels of OA (37 ng SPATT^−1^) and homo-YTX (15 ng SPATT^−1^) remained constant independently of the period. These findings suggest background contamination similar to that in a previous study conducted in French Polynesia at Nuku Hiva (a ciguateric hotspot) [41], where SPATTs deployed during the same periods showed similar contents of OA (around 40 ng SPATT^−1^), regardless of the season. Similarly, SPATT disks deployed in Nigerian waters, another tropical region, also showed low to intermediate concentrations of OA and PTX2 (up to ca. 180 and 120 ng SPATT^−1^, respectively), where only very low cell numbers of *Dinophysis* and *Prorocentrum* were detected [42]. In a study by Mackenzie et al., 2011 [43], SPATT bags were deployed in two warm, albeit temperate, estuaries of New Zealand (Parengarenga and Rangaunu Harbors, North Island) during two summer periods (February 2009 and 2010). In that study, intermediate to high levels of OA (132 to 2364 ng SPATT^−1^) and PnTXs (40 to 904 ng SPATT^−1^) were found, together with trace amounts of other lipophilic toxins, such as SPX1, PTX2, and DTX1, which is consistent with concomitantly observed blooms of *Dinophysis* in the area.

Considering PTX2, our results highlight an increase in PTX2 during the warm season (20 times higher). The observed enhancement in the SPATT samplers during the summer period suggested a higher concentration of phytoplankton primary producers (*Dinophysis* spp.), but the levels were still low compared with what we could expect during a bloom of *Dinophysis* in warm temperate regions (e.g., 421 ng PTX2 SPATT^−1^ [26], approximately 5500 ng PTX2 SPATT^−1^ [28]). Also, Zendong et al., 2016 [44], observed approximately a ten-fold increase in PTX2 in SPATT samples from one week to the next, suggesting that significant week-to-week variation occurs in coastal environments, including lagoons and embayments. The PTX2 detected in the present study could possibly either be adsorbed from the dissolved phase from a previous toxic event or be due to the presence of *Dinophysis* at variable, but low, concentrations throughout the year.

Similarly, AZA2 and PnTX-G could originate either from a previous bloom of species of the *Amphidomataceae* or *Vulcanodinium rugosum*, respectively, or originate from a background occurrence of these organisms in the area.

In the Yellow Sea, China [45], a one-year monthly monitoring campaign using SPATTs was carried out. They detected OA, DTX1, and PTX2 in seawater samples, with YTX also present at a trace level (<LOQ). These results showed the highest levels of toxins in June and August, peaking at 495 ng SPATT^−1^ OA, 168 ng SPATT^−1^ DTX1, and 375 ng SPATT^−1^ PTX2, and the lowest levels of toxins were measured in December (around 3 ng SPATT^−1^). As expected in temperate systems, seasonality had an impact on the toxins level, with a higher concentration during the summer (June and August for the northern hemisphere). Still, in a study conducted in the Yellow Sea [45], following the high concentrations found in June and August, the OA content decreased to 40 and 54 ng SPATT^−1^ and the PTX2 content decreased to 120 and 15 ng SPATT^−1^ in July and September, similar concentrations as the one quantified in our study. Therefore, our study likely missed the actual bloom event that led to an increase in the concentration of PTX2 in the warm season, and a higher toxin concentration in SPATTs might be expected in NC during a HAB event. To clarify this point in NC, SPATTs should be deployed monthly over a year-long period to link environmental data with contamination kinetics.

### 3.2. SPATTs and Bivalves

The major difference between toxin profiles in SPATTs and in filter-feeding bivalves is that toxins are metabolized and biotransformed once filtered by bivalves. In the bays surrounding Noumea, PTX2 was only detected in giant clams (*T. maxima*) and dissolved seawater (SPATTs), whereas PTX2sa was found in giant clams, mussels, and clams (*T. maxima, M.* cf. *auriculatus,* and *B. foliata,* respectively), but not in SPATTs. Suzuki et al. [46] showed that PTX2 is quickly biotransformed into PTX2sa in scallops. Similarly, Vale et al. [47] reported that PTX2sa was mainly found in shellfish flesh (e.g., mussels, cockles and oysters), whereas PTX2 was only detected in phytoplankton. These observations support the fact that exclusively PTX2 was found dissolved in seawater (i.e., in SPATTs) during the warm and cold periods, whereas. PTX2sa was not detected. Similarly, according to Ciminiello et al. [48], the major part of 45-OH-YTX, COOH-YTX, and its homo form are by-products of YTX and hYTX due to shellfish metabolism. In addition, Aesen et al. [49] reported that bivalve mollusks quickly oxidized YTX (or hYTX) to 45-OH YTX (or 45-OH hYTX) and more slowly to COOH YTX (or COOH-hYTX). These results are in accordance with the present study, as we found hYTX in both SPATT and bivalves, whereas we only detected its oxidized products (45-OH hYTX and COOH-hYTX) in bivalves. OA, PnTX-G and AZA2 were also detected in SPATTs as well as in shellfish. A major concern arises from the presence of PnTX-G in all shellfish collected and at higher levels than OA, while the amount of OA detected in SPATTs was greater. In a previous study, Mc Kenzie et al. [43], also found high levels of PnTX accumulated in oysters *C. gigas*, but without OA or DTXs, while those toxins were detected in SPATTs at significant levels. Conversely, all bivalves collected in bays surrounding Noumea accumulated as much GYM-A as PnTX-G, but GYM-A was not detected in SPATTs. These results raise the question of the ability of filter feeders to accumulate some dissolved marine biotoxins. Effectively, after a HAB event, toxins are released into the water and can be captured by SPATTs, but the ability of bivalve mollusks to accumulate some dissolved toxins more than others is debatable [50]. Indeed, Ji et al., 2020 [51], and Lamas et al., 2021 [52], showed the natural accumulation of GYM-A and esters of GYM-A in shellfish from Chinese and Spanish waters, respectively, and Pan et al., 2022 [53], showed that GYM-A can easily be absorbed by shellfish at low levels from the dissolved phase. However, dedicated studies need to be designed to observe a trend and clarify the links between the toxins detected in SPATTs and in bivalves [26].

### 3.3. Toxins and Risk for Population and Shellfish Industries

In parallel to shrimp farming, two oyster farms have been in place for more than fifteen years in NC. A diversification strategy, supported by local communities, has been launched that mainly concerns oyster, fish, sea cucumbers, and, to a lesser extent, other bivalves, such as giant clams and scallops, and mangrove crabs. Today, there is a lack of knowledge about HABs and their toxins in NCia; there is no monitoring program and almost no data on toxic microalgae and their associated toxins. In the present work, 11 groups of lipophilic toxins were screened, and of the 6 groups detected in the shellfish, 3 are regulated in the EU (OA/DTXs, YTXs and AZAs) and 1 in the US, Mexico, Australia, and New Zealand (BTX), with a regulatory threshold of 800 µg BTX2 kg^−1^ of shellfish flesh, identical to the internationally agreed limit in *Codex Alimentarius* Standard 292/2008 [3]. These results indicate that these toxins could have potential sanitary and socioeconomic impacts in NC.

Interestingly, dermatitis or itching were reported through an epidemiological survey of groups of swimmers in Lemon Bay during the same period of sampling in February 2022, but no reports were made in August 2021. As recently reported, the microalgae *Vulcanodinium rugosum* was implicated in several cases of major dermatitis in Cuba [54] and Senegal (P. Hess, personal communication). *Vulcanodinium rugosum* is the only known producers of PnTXs; this recent group of toxins, although not yet regulated in Europe or in the world, is controlled by the French National Sanitary Agency (ANSES), which recently recommended a concentration lower than 23 µg PnTX-G kg^−1^ of shellfish flesh [39]. PnTXs were apparently not related to the dermatitis reported by Moreira-González et al. [54], but the detection of this group of toxins suggests the presence of *V. rugosum*. In the present study, PnTX-G was found in all bivalves collected in bays surrounding Noumea. The highest level of PnTX-G was detected in *B. foliata* (22.6 µg PnTX-G kg^−1^), which was close to the threshold recommended by the ANSES. Other microalgae, such as *Ostreopsis* species have been associated with skin irritation [55]. Although, *Ostreopsis* spp. are present in NC [56], PLTX or its microalgal analogue ovatoxins or ostreocins were not found in either SPATTs or shellfish. While ovatoxin recovery from SPATTs is notoriously poor [57], significant levels of ovatoxins have been detected in mussels from affected areas [58].

Many regulated and unregulated biotoxins were detected in the present study, highlighting a potential risk to the consumers of filtering bivalves. However, the levels observed did not exceed the regulatory thresholds (Table 1), but, as mentioned, the two-week one-spot survey was obviously not designed to detect HAB events and only measured a basal concentration commonly found in filter-feeders in this tropical area. Still, the occurrence of multiple toxin groups in New Caledonian waters and shellfish clearly suggests that there is a potential risk. Thus, monthly or bi-weekly monitoring studies need to be conducted to assess the seasonality and to determine if high-risk periods with actual HAB events exist. However, even with small quantities, the accumulation of different groups of toxins with many different symptoms is a form of co-occurrence or “cocktail” that raises toxicological questions, especially in a region of the world where ciguatera is a major problem. For example, OA and AZA2, although only present at trace levels, can contribute to intestinal disorders often associated with ciguatera in NC.

This presence of internationally regulated toxins in the EU, US, Mexico, Australia, and New Zealand in the context of (i) the practice of local mollusk fisheries, (ii) at least two oyster farms producing for the local market and (iii) ongoing pilot studies for scallop fishing for exportation [17] all highlight a strong need for the establishment of a systematic monitoring program. 

## 4. Conclusions

This survey reports marine algal toxins in seawater (via passive sampling based on the SPATT approach) and bivalve mollusks in the bays surrounding Noumea. Several groups of regulated toxins were detected for the first time in NC, including okadaic acid (OA), azaspiracids (AZAs), and brevetoxins (BTXs). Furthermore, pectenotoxins (PTXs), yessotoxins (YTXs), and cyclic imines (GYMs, SPXs, and PnTXs, respectively) were detected, some analogs close to recommended safety thresholds (i.e., PnTX-G). The detection of such toxins in seawater or filter-feeding bivalves suggests that the primary producers of these toxins (listed in Table 1) are present in NC. The absence of data on toxic microalgae implies a lack of knowledge of the importance of HABs for the different professional and leisure uses of New Caledonian coastal ecosystems. In order to improve the knowledge on the potential impacts of HABs and their toxins on environmental and human safety, a more systematic study should be carried out on an extended area over a longer period and at more frequent intervals to assess the necessity of establishing a regular monitoring program in NC.

## 5. Materials and Methods

### 5.1. Study Area and Sampling Site 

NC is located in the Southwestern Pacific Ocean at approximately 1500 km east of Australia. NC is composed of one main island, Grande Terre, the Loyalty Islands, and dozens of smaller islands. Grande Terre is surrounded by a 1600 km long barrier reef, which delimits a 23,400 km^2^ lagoon [59].

Our study focuses on one main station in the south of Lemon Bay, the main beach of Noumea, the capital of NC, which is part of the Great South Lagoon. This sandy beach is surrounded on its north and south parts by two small degraded and anthropized coral reefs, in a shallow-water area (maximum depth ~1–4 m), and is exposed to urbanization (Figure 4B). The sampling devices were placed next to the coral reef (Figure 4B) on a sandy bottom area. The filter-feeding bivalves were then sampled close to the study site on the coral reef. The station was monitored during the cold (August 2021) and warm (February 2022) seasons over a week. The two other sites were sampled to collect filter-feeding bivalves during the warm season. They were situated on and next to a small degraded and anthropized reef, in a shallow-water area (maximum depth ~2 m), situated in the northwestern part of Vata Bay, and in a small intertidal area in the western part of Sainte Marie Bay (Figure 4B).

### 5.2. Solid Phase Adsorption Toxin Tracking (SPATT): Handling and Extraction Procedure

The SPATTs were prepared with 5 g of Diaion^®^ HP-20 polymeric resin (Sigma-Aldrich, Saint-Quentin-Fallavier, France) and were placed between sheets of nylon mesh (100 µm) and squeezed into an embroidery frame. The resin was activated overnight in 100% methanol (MeOH). The disk was gently washed with Milli-Q water to remove the MeOH-residue, and then soaked in Milli-Q water. After conditioning, it was stored in Milli-Q water at 4–6 °C for up to 2 days before use. Three SPATTs were deployed at 1.5 m from the sandy bottom, at 1.5 m from each other, and at around 2.5 m depth and left submerged for a week. The SPATTs were then recovered and cleaned at the laboratory using fresh water and conserved at −80 °C. 

Before extraction, the SPATTs were opened, and the resin was washed with 1 L of Milli-Q water, and placed in a glass column (3 × 60 cm) with a glass frit at the bottom. The toxins were then slowly eluted at 1 mL min^−1^ with three volumes of MeOH (10 mL). Subsequently, the combined extracts were evaporated using a rotary evaporator, and reconstituted in 1 mL of MeOH before LC-MS/MS analysis.

### 5.3. Shellfish Samples: Collection and Extraction Procedure

Shellfish were collected from three bays surrounding Noumea (Lemon, Vata and Sainte Marie Bays, Figure 4). Some of these species are often fished and eaten in New Caledonia (e.g., *Tridacna maxima, Pinctada* spp,. and *Anadara scapha*, Table 2). The bivalve mollusks were rinsed with Milli-Q water and the whole flesh of each species was extracted, pooled, and homogenized. All samples were freeze-dried before extraction. 

The lyophilized whole tissue (WT) of shellfishes was extracted twice using MeOH 90% (1:15, *w*/*v*) using an ULTRA TURRAX homogenizer (12,000 tr min^−1^, 5 min). After centrifugation (10 min at 3000 rpm), the supernatants were pooled and homogenized. An aliquot of the extract was filtered through 0.2 µm before being analyzed for multi-toxins by LC-MS/MS.

### 5.4. LC-MS/MS Multi-Toxin Analysis

Quantitative targeted analyses were conducted by LC-MS/MS on the methanolic extracts following methods described previously for most of the major lipophilic marine biotoxin groups (Table 2). 

In the present study, we only described in detail analytical methods 1 to 3 (Table 3), which allowed us to detect the lipophilic toxins present in SPATTs and in shellfish samples. To perform these methods, an Ultra-High-Performant Liquid Chromatographic system (UFLC XR Nexera, SHIMADZU, Tokyo, Japan) coupled to a hybrid triple quadrupole–linear ion trap API4000 QTRAP mass spectrometer (Sciex, Framingham, MA, USA) equipped with a TurboV^®^ electrospray ionization (ESI) source was used. The instrument control, data processing, and analysis were conducted using Analyst software 1.7.2 (Sciex, Framingham, MA, USA). A 2.6 µm C_18_ Kinetex column (50 × 2.1 mm, Phenomenex, Le Pecq, France) was employed at 40 °C and eluted at 400 µL min^−1^ with a linear gradient specific to each method. Eluent A was water and eluent B was 95% acetonitrile, both containing 2 mM ammonium formate and 50 mM formic acid. The acquisition was operated in the multi reaction monitoring (MRM) mode, scanning two or three transitions for each toxin.

#### 5.4.1. Method 1: Detection of OA, DTX and YTX Groups in ESI^-^

To detect the OA and YTX groups, the acquisition was carried out in the negative mode and the MRM transitions used are displayed in Appendix A. The conditions of the ESI^-^ source were set as follows: curtain gas (CUR) at 20 psi, ion spray (IS) at −4500 V, turbogas temperature (TEMP) at 550 °C, gas 1 and 2 (GS1 & GS2) at 40 and 50 psi, respectively, and an entrance potential (EP) of −13 V. Chromatographic separation was achieved using the following gradient: rising from 10% to 50% in 2 min, increasing to 95% in 4 min, held at 95% for 2 min before returning to the initial conditions in 0.5 min, and followed by an equilibration period of 3.5 min. Quantification was performed from linear calibration curves generated from a certified standard of OA, DTX2, DTX1, YTX, and Homo-YTX (NRC-CNRC, Halifax, NS, Canada). The total amount of OA and DTX toxins (fatty acyl esters, diol esters…) were not sought in this study as no hydrolysis step was followed.

#### 5.4.2. Method 2: Detection of AZAs, PTXs and Cyclic Imines in ESI^+^

The instrument was operated in the positive MRM mode the selected *m/z* transitions are reported in Appendix A. The source parameters used were: CUR at 30 psi, IS at 5500 V, GS1 and GS2 at 50 psi, TEMP at 450 °C, and EP at 10 V. The elution gradient started with 20% of B held for 0.5 min, then increased to 95% B in 5.5 min and held for 2 min before returning to the initial conditions in 0.5 min, followed by 3.5 min of equilibration.

#### 5.4.3. Method 3: Detection of BTXs in ESI^+^

To detect the BTX group, chemical analysis was carried out in the positive MRM mode and the chosen *m/z* values are reported in Appendix A. The gradient used was: 30–70% B for 1 min, 70–95% B for 9 min, 95–100% B for 0.1 min and maintained for 1.9 min, then 100–30% B for 0.1 min and held 3.9 min for equilibration. The following source settings were used: curtain gas at 20 psi, ion spray at 5500 V, turbo gas temperature of 300 °C, and gas 1 and 2 at 40 and 50 psi, respectively. Quantification was performed from linear calibration curves generated from certified standards of BTX1 to BTX3 and BTX-B5 (Novakits, Nantes, France).

## Figures and Tables

**Figure 1 toxins-15-00642-f001:**
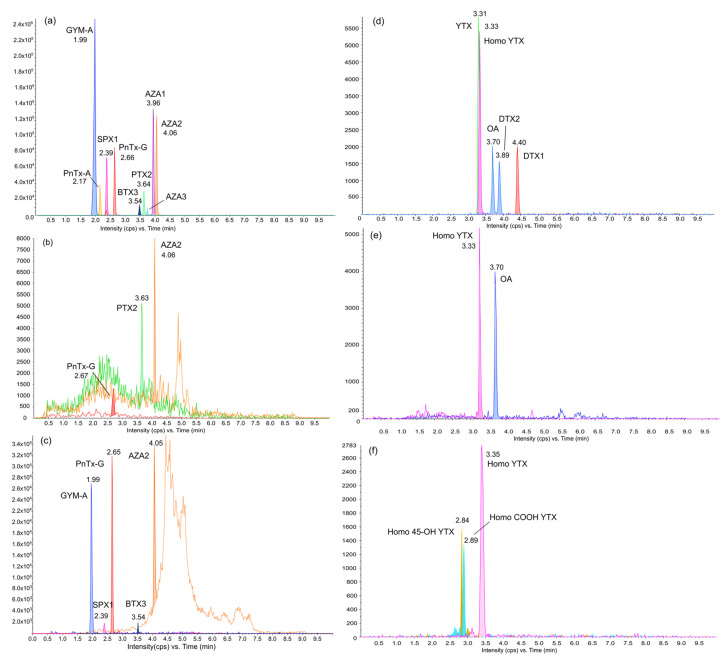
LC-MS/MS chromatogram acquired in ESI^+^ (**a**–**c**) and in ESI^−^ (**d**–**f**) representing the most intense MRM transition for each compound. Chromatograms (**a**,**d**) represent a mixture of certified standards (NRC-CNRC, Halifax, Canada); (**b**,**e**) an extract from the SPATT sample; (**c**,**f**) an extract from the whole tissue of *Pteria straminae*.

**Figure 2 toxins-15-00642-f002:**
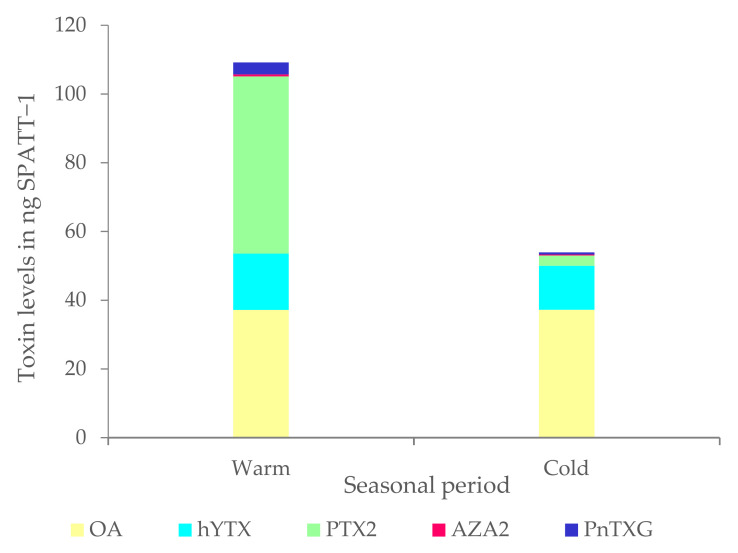
Toxin levels (mean, *n* = 3) accumulated in SPATTs deployed for one week in Lemon Bay, Noumea, at two seasonal periods.

**Figure 3 toxins-15-00642-f003:**
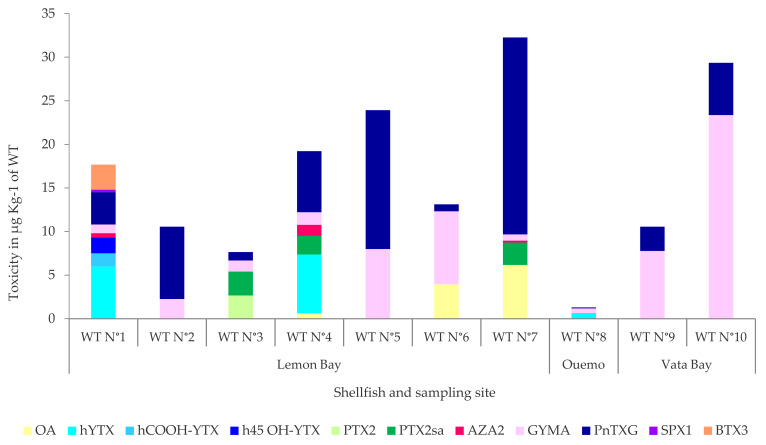
Distribution of lipophilic toxins detected by LC-MS/MS in a wide range of shellfish (Table 2) species (Whole tissue, WT) collected in the bays of Noumea in the order of: (N°1) *Pteria straminea*, (N°2) *Pinctada maculata*, (N°3) *Tridacna maxima*, (N°4) *Modiolus* cf. *auriculatus*, (N°5–7) *Barbatia foliata*, (N°8) *Pinctada margaritifera*, (N°9) *Anadara scapha or* cf. *trapezia*, and (N°10) *Isognomon isognomon*.

**Figure 4 toxins-15-00642-f004:**
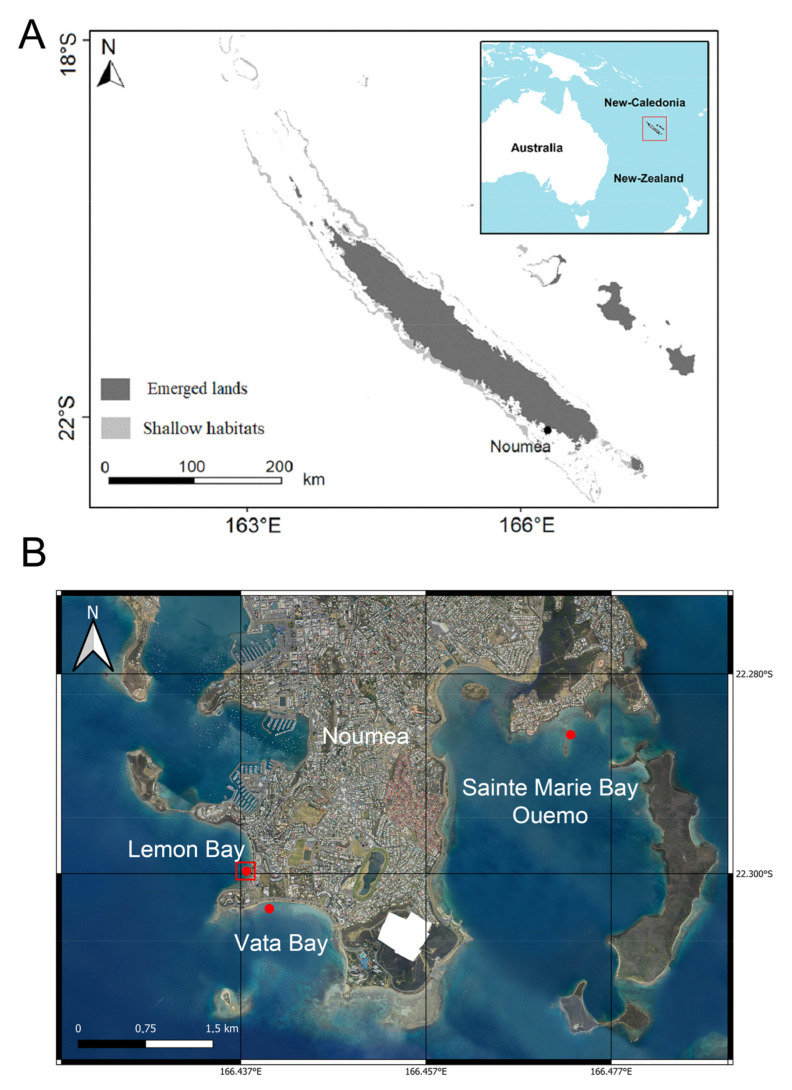
Study site. (**A**) Location of Noumea in New Caledonia. (**B**) Satellite view of Noumea’s Bays, highlighting the different sampling stations (red points). The red box is the place where the passive samplers (SPATTs) were deployed.

**Table 2 toxins-15-00642-t002:** List of the bivalve mollusks collected in the bays of Noumea.

Code	Species	Type	Sampling Site	Sampling Period	*n*	Tissue
WT N°1	*Pteria straminea*	/	Lemon Bay	Cold	5	Whole flesh
WT N°2	*Pinctada maculata*	Oyster	Lemon Bay	Warm	3	Whole flesh
WT N°3	*Tridacna maxima*	Giant clam	Lemon Bay	Warm	1	Whole flesh
WT N°4	*Modiolus* cf. *auriculatus*	Mussel	Lemon Bay	Warm	6	Whole flesh
WT N°5	*Barbatia foliata*	Clam	Lemon Bay	Warm	1	Whole flesh
WT N°6	*Barbatia foliata*	Clam	Lemon Bay	Warm	1	Whole flesh
WT N°7	*Barbatia foliata*	Clam	Lemon Bay	Warm	1	Whole flesh
WT N°8	*Pinctada margaritifera*	Oyster	Ouemo	Warm	4	Whole flesh
WT N°9	*Anadara scapha or* cf. *trapezia*	Cockle	Vata Bay	Warm	5	Whole flesh
WT N°10	*Isognomon isognomon*	Oyster	Vata Bay	Warm	10	Whole flesh

**Table 3 toxins-15-00642-t003:** List of the targeted LC-MS/MS methods applied to methanolic extract of SPATTs and bivalve mollusks.

LC-MS/MS	References	Toxins Group	Abbreviation	Targeted Toxins
**Method 1**	Zendong et al., 2017 [60] with modifications	Okadaïc acid	OA/DTXs	OA, DTX1, and DTX2
		Yessotoxines	YTXs	YTX, homo YTX, 45-OH YTX, homo 45-OH YTX, COOH YTX, and Homo COOH YTX
**Method 2**	Zendong et al., 2017 [60] with modifications	Azaspiracids	AZAs	AZA1 to 3
		Pectenotoxins	PTXs	PTX1, PTX2, PTX11, and PTX2sa
		Cyclic imines	GYMs	Gymnodimine A, B, and C
			SPXs	SPX-A to -D, and SPX desMe-C and -D
			PnTXs	PnTX-A to -H
			PtTX	Pte-A to -C
				Portimine
**Method 3**	This study	Brevetoxins	BTXs	BTX1 to 3, BTX-B5, BTX6, BTX7, and BTX9 and Brevenal
Method 4	Yon et al., 2021b [61]	Maitotoxines	MTXs	MTX1, MTX2, and MTX4 to MTX7, desulfo-MTX1, and desulfo-MTX2
		Gambiertoxines		Gambierone, 44-Me Gambierone (ex MTX3), sulfo-gambierone, dihydro-sulfo gambierone gambieroxide, and gambieric acid -A to -D
Method 5	Estevez et al., 2020 [62]	Ciguatoxines	C-CTXs	C-CTX1 to 4
			I-CTXS	I-CTX1 to 6
Method 6	Sibat et al., 2018 [63]		P-CTXs	Type CTX3C and type CTX1B (>20 compounds)
Method 7	Chomerat et al., 2019 [64]	PLTX-like	PLTXs	PLTX and 42-OH PLTX
			OvTXs	Ovatoxines a to k
			MscTXs	Mascarenotoxins A to C
			OSTs	Ostreocin-A, -B, -D, and -E1
Method 8	Georges des Aulnois et al., 2019 [65]	Microcystins	MCs	dmMC-RR, MC-RR, MC-YR, MC-LR, dm-MC-LR, MC-LA, MC-LY, MC-LW, MC-LF, and nodularin (NOD)

## Data Availability

Not applicable.

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
