# Peer review of "Seasonal Single-Site Sampling Reveals Large Diversity of Marine Algal Toxins in Coastal Waters and Shellfish of New Caledonia (Southwestern Pacific)"

_toxins, 2023, doi:10.3390/toxins15110642_

Round 1
Reviewer 1 Report
Comments and Suggestions for Authors
This manuscript reports results of the survey of several algal toxins in coastal seawater and shellfish of New Caledonia. Okadaic acid (OA), azaspiracid-2 (AZA2), pectenotoxin-2 (PTX2), pinnatoxin-G (PnTX-G) and homo-yessotoxin 11 (homo-YTX) were detected in seawater at higher levels during summer. A more diversified toxin was found in shellfish with brevetoxin-3 (BTX3), gymnodimine-A (GYM-A), and 13desmethyl spirolide-C (SPX1). This is the first report of the toxin profiles in seawater and shellfish in New Caledonia. Because the data also seems solid, I think this paper is worth publishing in this Jouanal.
One thing, I am concerned that in Table 1, the species involved in tetrodotoxin was described as Tetraodontinae, Prorocentrum minimum. I think that Prorocentrum minimum has been proposed, but not clearly determined. So I suppose it is too early to describe it in this Table.
Author Response
We gratefully thank the reviewer 1 to approve our work and the time he provided to review our manuscript. Concerning his comment and to avoid confusion, Prorocentrum minimum was removed from Table 1 in the revised manuscript.
Reviewer 2 Report
Comments and Suggestions for Authors
Line 63 - …, Pilot… Most likely, the word should begin with a small letter
Line 67 - LC-MS/MS Please decipher this abbreviation.
Line 105 - Table 1 It seems to me that it would be better to move the table higher and insert it after the first mention in the text.
Line 106 – There is no need to highlight references in bold in the column Regulatory limit (in bold)…
Table 1 - DSP1, NSP2, ASP3 , PSP4 What do these numbers mean?
Table 1 - Regulatory limit or recommended threshold value for Cyanotoxins, Microcystins have no references.
Line 125 – CV Please decipher this abbreviation in text.
Line 156 - I would recommend presenting the results in the form of a heatmap with UPGMA analysis for these samples.
Line 290 - In this case it is better to start with "For example"
Line 398 - Table S4 There is no such table number in the supplementary materials, please check. There is also no reference in the text to tables (Table S2, Table S3). Also check this point.
Line 355 - How are the samples different WT N°5, WT N°6, WT N°7? They have the same parameters in the table.
Table S2 - I recommend highlighting in bold where acceptable values were observed to be exceeded.
Line 409 - Please check the list. The authors' surnames are only abbreviated in several places. For example, A.-A.N.Z.F. line 470
In general, the work is understandable and very relevant today. Although few sampling points were chosen and only two seasons were studied, the authors suggest further work. A comparative analysis of toxins in cold and warm times, it seems to me, requires a larger number of samples and a wider sampling area. I think future studies will accomplish this. Thus, this work will allow the authors to further conduct both a comparative analysis and serve as a foundation for further research.
Has a control point been sampled for seawater analysis? Also, is it possible to sample shellfish in northern New Caledonia as a control point?
Reviewer 3 Report
Comments and Suggestions for Authors
The research contributes to knowledge of toxin distribution in the SW Pacific area.
The ms requires many improvements before acceptance as outlined below. These are mostly related to an improper review of the manuscript before submission.
Line 49- the abbreviation ‘NC’ was not introduced so far to the reader (it comes only later in line 92)
Table 1 - contains several footnotes (1-4), but these were not explained in the end of the table.
Table 1 – intoxication by TTXs is not called ‘PSP’. PSP is a distinct syndrome. Its best to use TTX poisoning.
Line 111- Indication of figs (c) and (f) are missing, as the text mentions both spatt and shellfish data.
Line 122 – the abbreviation for ‘HP’ was not presented. Also, the use of hepatopancreas was not discriminated in the method’s section.
Line 136 – the results state that PnTX-G was present in all shellfish samples. But this toxin was not included in the legend of Fig. 3, nor reported in the bars.
Line 265 – who made the personal communication? From which institution?
Lines 323-326- Fig. 1A does not illustrate the nearby urbanization, only the global localization of the study area.
Line 326 – Fig. 1C is non-existent.
Section 5.4.1 – the method used for okadaites is appropriate for SPATT but not for shellfish. Many bivalve’s bio transform okadaites into acyl esters, which cannot be detected by screening simply for okadaic acid. The hydrolysis step was not included here, and the authors should mention this underestimation in toxicity (in addition to the sampling bias of the single-site sampling)
Line 398 – Table 4 does not appear in the supplementary file I received.
Comments on the Quality of English Language
none
